# LOCAL DISTANCE PRESERVING AUTO-ENCODERS US-ING CONTINUOUS K-NEAREST NEIGHBOURS GRAPHS

## ABSTRACT

Auto-encoder models that preserve similarities in the data are a popular tool in representation learning. In this paper we introduce several auto-encoder models that preserve local distances when mapping from the data space to the latent space. We use a local distance-preserving loss that is based on the continuous k-nearest neighbours graph which is known to capture topological features at all scales simultaneously. To improve training performance, we formulate learning as a constraint optimisation problem with local distance preservation as the main objective and reconstruction accuracy as a constraint. We generalise this approach to hierarchical variational auto-encoders thus learning generative models with geometrically consistent latent and data spaces. Our method provides state-of-the-art or comparable performance across several standard datasets and evaluation metrics.

## 1 INTRODUCTION

Auto-encoders and variational auto-encoders (Kingma & Welling, 2014; Rezende et al., 2014) are often used in machine learning to find meaningful latent representations of the data. What constitutes meaningful usually depends on the application and on the downstream tasks, for example, finding representations that have important factors of variations in the data (disentanglement) (Higgins et al., 2017; Chen et al., 2018), have high mutual information with the data (Chen et al., 2016), or show clustering behaviour w.r.t. some criteria (van der Maaten & Hinton, 2008). These representations are usually incentivised by regularisers or architectural/structural choices.

One criterion for finding a meaningful latent representation is geometric faithfulness to the data. This is important for data visualisation or further downstream tasks that involve geometric algorithms such as clustering or kNN classification. The data often lies in a small, sparse, low-dimensional manifold in the space it inhabits and finding a lower dimensional projection that is geometrically faithful to it can help not only in visualisation and interpretability but also in predictive performance and robustness (e.g. Karl et al., 2017; Klushyn et al., 2021). There are several approaches that implement such projections, ISOMAP (Tenenbaum et al., 2000), LLE (Roweis & Saul, 2000), SNE/t-SNE (Hinton & Roweis, 2002; van der Maaten & Hinton, 2008; Graving & Couzin, 2020) and UMAP (McInnes et al., 2018; Sainburg et al., 2021) aim to preserve the local neighbourhood structure while topological auto-encoders (Moor et al., 2020), witness auto-encoders (Schönenberger et al., 2020), and (Li et al., 2021) use regularisers in auto-encoder models to learn projections that preserve topological features or local distances.

The approach presented in (Moor et al., 2020), uses persistent homology computation to define local connectivity graphs over which to preserve local distances. One can choose the dimensionality of the preserved topological features, however, preserving higher-dimensional topological features comes at additional computational cost. In this paper we propose to use the continuous k-nearest neighbours method (Berry & Sauer, 2019) which is based on consistent homology and results in a significantly simpler graph construction method; it is also known to capture topological features at all scales simultaneously. Since AE and VAE methods are usually hard to train and regularise (Alemi et al., 2018; Higgins et al., 2017; Zhao et al., 2018; Rezende & Viola, 2018), to improve learning we formulate learning as a constraint optimisation with the topological loss as the objective the reconstruction loss as constraint. In addition, we adapt the proposed methods to VAEs with

learned priors. This enables us to learn models that generate data with topologically/geometrically consistent latent and data spaces.

## 2 METHODS

In this paper we address (i) projecting i.i.d. data $X = \{x_i\}_{i=1}^N$ with $x \in \mathrm{R}^n$ into a lower-dimensional representation $z \in \mathrm{R}^m$ ($m < n$) using auto-encoders and (ii) learning an unsupervised (hierarchical) probabilistic model that can be used not only to encode but also generate data similar to $X$. Auto-encoder models are typically learned by minimising the average reconstruction loss $L_{\mathrm{rec}}(\theta, \phi; X) = \mathrm{E}_{\hat{p}(x)}[l(x, g_\theta(f_\phi(x))]$ w.r.t. $(\theta, \phi)$, where $l(\cdot, \cdot)$ is a positive, symmetric, non-decreasing function and the mappings $f_\phi$ and $g_\theta$ are called the encoder and the generator, respectively. Due to consistency with distance preserving losses, we only use as reconstruction loss the Euclidean distance $l(x, x') = ||x - x'||^2$. The expectation is taken w.r.t. the empirical distribution $\hat{p}(x) = (1/N) \sum_i \delta(x - x_i)$ and training is performed via stochastic batch gradient methods.

Unsupervised probabilistic models are typically learned by maximum likelihood method w.r.t. $\theta$ on $p_\theta(X) = \prod_i \int_i p_\theta(x_i|z_i) p_\theta(z_i) \, dz_i$, where $p_\theta(x|z)$ is the likelihood term corresponding to the generator $g_\theta(x)$ and $p_\theta(z)$ is the prior distribution/density of the latent variables $z$. The distribution $p_\theta(z)$ is either chosen as a product of some standard univariate distributions or learned via empirical Bayes. In practice, learning the prior is often included in the maximum likelihood optimisation. Since the integrals $\int_i p_\theta(x_i|z) p_\theta(z) dz$ are usually intractable, $\log p_\theta(x)$ is often approximated using amortised variational Bayes (Kingma & Welling, 2014; Rezende et al., 2014) resulting in the evidence lower-bound (ELBO) approximation $\log p_\theta(x) \geq \max_\phi \left\{ \mathrm{E}_{q_\phi(z;x)}[\log p_\theta(x|z)] - \mathrm{KL}[q_\phi(z;x)||p_\theta(z)] \right\}$. The resulting $q_\phi(z; x)$ is an approximation of the posterior distribution $p_\theta(z|x) = p_\theta(x|z) p_\theta(z)/p_\theta(x)$ and can be viewed as corresponding to the encoder $f_\theta(x)$. For notation simplicity, we use $\theta$ for all model parameters, and $\phi$ for all encoder parameters. In this paper we will deviate slightly from the ELBO approach to fit the parameters $\theta$ and $\phi$ because of practical considerations but the general modelling ideas will be similar nonetheless.

### 2.1 LOCAL DISTANCE PRESERVATION

Auto-encoders are popular models for dimensionality reduction and thus they are often extended with regularisers or constraints that impose various types of inductive biases required by the task at hand. One such inductive bias is local distance preservation, that is, two data points $x_i$ and $x_j$ close in the data-space at distance $d_\mathcal{X}(x_i, x_j)$ should be mapped into points $z_i = f_\phi(x_i)$ and $z_j = f_\phi(x_j)$ at distance $\gamma d_\mathcal{Z}(z_i, z_j) \simeq d_\mathcal{X}(x_i, x_j)$. This distance preservation can help to retain the topology of the data $X$ in the encoded data $Z = \{z_i = f_\theta(x_i)\}_{i=1}^N$. Since the the data $X$ is often hypothetised to lie on a sub-manifold of $\mathbb{R}^n$, give or take some observation noise (Rifai et al., 2011), one expects that the encoded data $Z$ will be a lower-dimensional, topologically faithful representation of $X$.

In this paper we mainly consider local distance preservation where locality or closeness in the data manifold is formulated via (neighbourhood) graph structures constructed based on topological/geometrical considerations. We present the graph construction methods we use in Section 2.3. Let us assume that we have constructed two graphs with the same method, a graph $\mathcal{G}_X$ based on data/batch and another graph $\mathcal{G}_Z$ based on the encoding of the data/batch. Given these graphs and the distance measures in both spaces, we define the local distance-preserving loss defined similarly as in (Sammon, 1969; Lawrence & Quinonero-Candela, 2006; Moor et al., 2020),

$$L_{\mathrm{topo}}(\phi; X, Z) = \sum_{(i,j) \in \mathcal{G}_X \cup \mathcal{G}_Z} |d_\mathcal{X}(x_i, x_j) - \gamma d_\mathcal{Z}(z_i, z_j)|^2. \tag{1}$$

Here, in case of auto-encoder models we have $Z = \{z_i = f_\phi(x_i)\}_{i=1}^N$, while in case of generative models we have $Z = \{z_i \sim q(z; x_i)\}_{i=1}^N$. The scaling factor $\gamma$ is a learned variable and is introduced to help with the scaling issues one might encounter in VAE models. In case of generative models one can also consider the generative counterpart for $Z' \sim p_\theta(z), X' \sim p_\theta(\cdot|Z')$. For models and training schedules we considering in this paper this did not bring any additional benefit because a good auto-encoding and a well fitted prior already ensures a small value for this additional term.

There are several other options for loss functions that are designed to incentivise auto-encoders to preserve locality structures. SNE/tSNE construct a probability distribution of connectedness for each

data point both in the data and latent spaces and compare these using the Kullback-Leibler divergence. UMAP uses a formally similar method on a symmetrised k-nearest neighbours graph (see Section 2.3) albeit based on different theoretical considerations.

## 2.2 INFERENCE AND LEARNING VIA CONSTRAINED OPTIMISATION

Probabilistic generative models (VAEs) are often hard to train because they can converge to sub-optimal local minima (Sønderby et al., 2016), moreover, it has been shown in several papers that higher ELBO values do not necessarily correspond to better prediction performance or informative latent spaces (Alemi et al., 2018; Higgins et al., 2017). For this reason, several annealing schemes have been proposed that slowly "turn on" the KL-divergence term in the ELBO to avoid an over-regularisation of $q_\phi$. In particular, scheduling schemes derived from constrained optimisation approaches (Rezende & Viola, 2018) can significantly improve training in hierarchical generative models (Klushyn et al., 2019). For this reason, we propose two constrained optimisation methods to train auto-encoders and generative models.

In case of auto-encoders, we formulate the optimisation problem as

$$\min_{\theta,\phi} \mathrm{E}_{X_b \sim \hat{p}(x)} \left[ L_{\mathrm{topo}}(\phi; X_b, f_\phi(X_b)) \right] \tag{2a}$$

$$\mathrm{s.t.} \ \mathrm{E}_{X_b \sim \hat{p}(x)} \left[ l\left( X_b, g_\theta\left( f_\phi\left( X_b \right) \right) \right) \right] \leq \xi_{\mathrm{rec}}, \tag{2b}$$

where $\xi_0^{\mathrm{rec}}$ denotes a baseline reconstruction error, a hyper-paramater that is mostly influenced by the model architecture. To emphasise that we use batch training and that $L_{\mathrm{topo}}$ is computed on a pair of data batch $X_b$, we overload the notation of the respective mapping and densities with this set notation.

In case of (hierarchical) generative models, we formulate the constrained optimisation problem

$$\min_{\theta,\phi} \mathrm{E}_{X_b \sim \hat{p}(x)} \left[ \mathrm{KL}[q_\phi(Z_b; X_b) || \, p_\theta(Z_b)] \right] \tag{3a}$$

$$\mathrm{s.t.} \ \mathrm{E}_{X_b \sim \hat{p}(x)} \left[ \mathrm{E}_{Z_b \sim q_\phi(\cdot; X_b)}[-\log p_\theta(X_b|Z_b)] \right] \leq \xi_{\mathrm{rec}} \tag{3b}$$

$$\mathrm{E}_{X_b \sim \hat{p}(x)} \left[ \mathrm{E}_{Z_b \sim q_\phi(\cdot; X_b)} \left[ L_{\mathrm{topo}}(\phi; X_b, Z_b) \right] \right] \leq \xi_{\mathrm{topo}}, \tag{3c}$$

where, when a Gaussian $p_\theta(x|z) = \mathcal{N}(x|g_\theta(z), \sigma_x^2)$ is used, we replace equation 3b with an equivalent reconstruction constraint $\mathrm{E}_{X_b \sim \hat{p}(x)} \left[ \mathrm{E}_{Z_b \sim q_\phi(\cdot; X_b)}[||X_b - g_\theta(Z_b)||^2] \right] \leq \xi_{\mathrm{rec}}$. The optimal parameter $\sigma_x^2$ can be computed at the end of training as the average square reconstruction error. The KL is overloaded to represent averaging over $X_b, Z_b$. The Lagrangian of the optimisation problem (3a–3c) has a similar form as an ELBO objective and thus resembles models in (Rezende & Viola, 2018), (Higgins et al., 2017) and (Klushyn et al., 2019) albeit with two constraint terms. In our experience the constraint optimisation approach leads to better training performance than simple regularisation when one has to fit objectives with different scales. In principle any of the three losses or weighted combinations of some/all can be considered as the main objective. To have automatic "weight tuning" via Lagrange multipliers it is reasonable to have one loss as objective and the rest as constraints. We have made the above choices, because it is relatively easy to come up with constraint boundary candidates: for the topological one we are informed by the distances in the data space, while for the reconstruction one by the reconstruction error.

To solve the optimisation problems (2a–2b) and (3a–3c), we define the corresponding Lagrangians and optimise them via gradient quasi-ascent-descent. We use the exponential method of multipliers (Bertsekas, 2003) for the Lagrange multipliers $\lambda_{\mathrm{rec}}$ and $\lambda_{\mathrm{topo}}$ corresponding to (2b,3b) and equation 3c, correspondingly. This reads as $\lambda_{\mathrm{rec}}^{t+1} = \lambda_{\mathrm{rec}}^t \exp\{\eta_{\mathrm{rec}}(\bar{L}_{\mathrm{rec}}^t - \xi_{\mathrm{rec}})\}$ and $\lambda_{\mathrm{topo}}^{t+1} = \lambda_{\mathrm{topo}}^t \exp\{\eta_{\mathrm{topo}}(\bar{L}_{\mathrm{topo}}^t - \xi_{\mathrm{topo}})\}$, where we use a first order moving averages $\bar{L}_{\mathrm{rec}}^t$ and $\bar{L}_{\mathrm{topo}}^t$ to dampen fast variations due to batch training (Rezende & Viola, 2018). There are several other options to fit $\lambda_{\mathrm{rec}}, \lambda_{\mathrm{topo}}$ such as various gradient methods on their logs. In addition we use the following simple tricks to maintain numerical stability: (i) we clip the multipliers at $10^2$–$10^4$ (ii) we set the objectives to $0$ until all constraints are first satisfied. The pseudocode of the training algorithm can be found in Algorithm 1.

## 2.3 BACKGROUND

In this section we present the local distance and/or topology preserving graph construction methods and losses we propose and compare to.

**Stochastic neighbourhood embedding (SNE/tSNE)** Instead of preserving topological structures, SNE/tSNE (Hinton & Roweis, 2002; van der Maaten & Hinton, 2008) proposes to preserve a distribution of distances/similarities for each data point $x_i$ and its encoding $z_i$ w.r.t. all/some other data points and encodings, respectively. This formulation allows the authors to use multi-modal encodings, however, in most applications SNE/tSNE is still used with a unimodal encoding.

For each data point $x_i \in X_b$, SNE/tSNE defines the probability of $x_j$ being a potential neighbour of $x_j$ as $p_{j|i}^X = k(x_i, x_j) / (\sum_{j \in \mathcal{N}(i)} k(x_i, x_j))$, where $k(\cdot, \cdot)$ is some distance or dissimilarity based kernel function, $\mathcal{N}(i)$ a set or possible neighbours according to some neighbourhood graph $\mathcal{G}$. In this paper we use fully connected graphs. Although several methods using sparse graphs have been developed for large datasets, using a full matrix is feasible in a stochastic batch gradient setting. To compute the probabilities we use the Student/Cauchy kernel $k(x_i, x_j) = 1/(1 + \delta^{-2}||x_i - x_j||^2)$ proposed in (van der Maaten & Hinton, 2008). The probability distributions in the latent space are defined similarly $p_{ij}^Z(\phi) = k(z_i, z_j)/(\sum_{j \in \mathcal{N}(i)} k(z_i, z_j))$, where, in case of auto-encoder models we have $z_i = f(x_i), z_j = f(x_i)$, while in case of generative models we have $z_i \sim q(z; x_i), z_j \sim q(z; x_j)$. Unlike in (van der Maaten & Hinton, 2008), based on practical considerations, here we the use symmetrised KL instead of symmetrised probabilities, and define the loss as $L(\phi; X_b, Z_b) = \frac{1}{2} \sum_i (\text{KL}[p_{\cdot|i}^X || p_{\cdot|i}^Z(\phi)] + \text{KL}[p_{\cdot|i}^Z(\phi) || p_{\cdot|i}^X])$.

**Uniform manifold approximation and projection (UMAP)** UMAP (McInnes et al., 2018; Sainburg et al., 2021) follows a similar approach as SNE/tSNE in the sense that it constructs a weighted sparse graph and defines a corresponding cross entropy based loss between the weights corresponding to the data $X_b$ and its encoding $Z_b$. The cross-entropy is computed via negative sampling and it only takes into account the graph constructed based on the data $X_b$. The authors prove that their weighted graph captures the underlying geometric structure of the data in a faithful way by using concepts from category theoretic approaches to geometric realisation of fuzzy simplicial sets (Spivak, 2009). The graph in the data space is constructed using a symmetrised weighted kNN graph. UMAP assigns the weights $w_{ij} = \alpha_i \exp\{-\max(0, d(x_i, x_j) - \min_j d(x_i, x_j))\}$ where the scaling $\alpha_i$ is defined such that $\sum_j w_{ij} = \log_2(k)$ in a kNN graph. This weight matrix is then symmetrised according to $\hat{w} = w_{ij} + w_{ji} - w_{ij} w_{ji}$. The weights for the encodings $Z_b$ are then computed similarly, albeit using $w_{ij} = 1/(1 + a||z_i - z_j||^{2b})$ with $a, b$ fitted based on theoretical assumptions. Due to the special edge-based batching schedule and loss computation of the parametric UMAP method in (Sainburg et al., 2021), we did not implement this method but used the open-source implementation instead.

**Vietoris–Rips complex (VR)** Moor et al. (2020) propose the regulariser in equation 1 for an auto-encoder model. The graph construction they propose is based on persistent homologies of Vietoris–Rips complexes (VR). A VR complex $\mathcal{R}_\epsilon(X_b)$ associated with the data points in $X_b$ at length scale $\epsilon$ is the set of all fully-connected components of the graph constructed based on pairwise $\epsilon$-ball connectivity. As $\epsilon$ increases the set $\mathcal{R}_\epsilon(X_b)$ contains more and more fully-connected components saturating when finally the whole graph is included. The authors apply persistent homology calculation on $\mathcal{R}_\epsilon(X_b)$ to obtain persistence diagrams and persistent pairings based on which one can identify simplices that create or destroy topological features.

It is shown that the for 0-dimensional topological features (connected components) the minimum spanning tree corresponding to the data $X_b$ and distance measure $d_\mathcal{X}$ contains all the topologically relevant edges. The authors show that their method works for higher-dimensional topological features (e.g. cycles, voids) but opt to use only 0-dimensional topological features and thus define the graphs $\mathcal{G}_{X_b}$ and $\mathcal{G}_{Z_b}$ as the corresponding minimum spanning trees. We used their publicly available implementation compute these graphs. The method in (Moor et al., 2020) provides a principled way to define a loss/regulariser that incentivises a topologically faithful encoding of the data together with a choice of complexity (dimension of topological features).

## 2.4 Continuous k-nearest neighbours (CkNN)

In contrast to *persistent* homology where different topological features arise at different length parameters $\epsilon$, Berry & Sauer (2019) propose *consistent* homology showing that it is possible to construct a single unweighted graph from which all topological information of the underlying manifold can be extracted. They propose the continuous k-nearest neighbours graph (CkNN), a graph that captures topological features at multiple scales simultaneously. They prove that it the unique unweighted graph construction for which the graph Laplacian converges spectrally to a

Laplace-Beltrami operator on the manifold in the large data limit. The method is applied to clustering and image pattern detection via PCA. To the best of our knowledge we are the first to adapt it to the context of deep generative models and stochastic batch gradient learning.

Let $\kappa(x; k, X_b)$ be the index of the k-th nearest neighbour of $x$ in $X_b$. Then the CkNN graph over the set $X_b$ is defined via the connectivity (Berry & Sauer, 2019)

$$\mathcal{G}_{X_b}(\delta, k) = \left\{ (i, j) : d_X(x_i, x_j)^2 \leq \delta^2 \, d(x_i, x_{\kappa(x_i; k, X_b)}) \, d(x_j, x_{\kappa(x_j; k, X_b)}), \ \ x_i, x_j \in X_b \right\}.$$

In other words, for $\delta = 1$, two points are connected if their distance is smaller than the geometric mean of they kNN radius/distance. Using a kNN-based approach has the benefit that it takes into account the local density of the points instead of the $\epsilon$-ball approach that works well only for data uniformly distributed on the manifold. In fact it is known for kNN that $||x - x_{\kappa(x; k, X_b)}|| \propto p(x)^{-1/m}$, where $p(x)$ is the sampling density and $m$ is the intrinsic dimension of the data.

As a result, CkNN is an instance of a broader class of graph constructions for where connectivity is defined by $d(x, x') < \delta[p(x)p(x')]^{-1/2m}$ and has the advantage that one does not have to estimate $m$, see (Berry & Sauer, 2019) for further details. This connection is specially interesting in the context of generative models where we learn the latent space and data distributions $p_\theta(z)$ and $p_\theta(x)$, respectively.

An additional advantage of CkNN is that it seems to be significantly faster to compute when compared to VR. For VR the computation of the minimum spanning tree is required on the (full) distance matrix resulting in $\mathcal{O}(n_{\text{batch}}^2 \log n_{\text{batch}})$ while for CkNN we only need to compute the smallest $k$ distances for each node, this is only $\mathcal{O}(n_{\text{batch}}^2 \log k)$ and is also highly paralleliseable. In Figure 1 we show how the wall-clock speeds of the latter two operations compare w.r.t. batch size.

## 2.5 LEARNING THE PRIOR

To define the prior models $p_\theta(z)$ we consider several known approaches with different degree of complexity and computational cost.

**The realNVP prior** The computationally simplest way to model a prior is to define the latent variable $z$ as an invertible transformation $z = h(\epsilon)$ of a factorising Gaussian or uniform variable $\epsilon \in \mathbb{R}^m$. This allows us to compute $\log p_\theta(z) = \log p_0(\epsilon(z)) + \log |\det(\partial \epsilon(z)/\partial z)|$ and therefore to approximate the KL divergence in equation 3a using a few Monte Carlo samples. Dinh et al. (2017) define $z = h(\epsilon)$ as a sequence of $K$ invertible transformations $z_{k+1}^{1:d} = z_k^{1:d}, z_{k+1}^{d+1:m} = z_k^{d+1:m} \odot \exp(s(z_k^{1:d}) + t(z_k^{1:d})), (d < m)$ with lower-triangular $\partial \epsilon(z)/\partial z$ and thus $\log p_\theta(z)$ can be computed efficiently. Note that $z$ and $\epsilon$ need to have the same dimensionality, therefore, in order for the computations to behave well, one should ideally chose latent dimensions $m$ for which the encoded data does not need to further projection to a lower-dimensional manifold.

**The VAMP prior** Tomczak & Welling (2018) define a learned prior in a VAE model starting from the observation that the optimal empirical Bayes prior is $p^*(z) = \mathbb{E}_{\hat{p}(x)}[q_\theta(z; x)]$, which holds for our objective in equation 3a as well. Based on this observation they propose the prior $p_\theta(z) = \sum_k q_\theta(z; y_k)/K$ with $K$ learnable pseudo-data parameters $y_1, \ldots, y_K$. This approach allows us to compute $\log p_\theta(z)$ efficiently and thus, to approximate the KL-divergence via sampling as mentioned above. The disadvantage of this definition is that we can only learn priors that can be well modelled with a few elliptical components.

**The hierarchical prior VHP** A more general approach to learning the prior is to use another hierarchy to model it (Klushyn et al., 2019), that is, to use $p_\theta(z) = \int p_\theta(z|\epsilon) \, p_0(\epsilon) \, d\epsilon$. This makes $\log p_\theta(z)$ intractable, however, we can further approximate it by using an importance-weighted bound (Burda et al., 2016) on $p_\theta(z)$ like in (Klushyn et al., 2019). This results in replacing KL-divergence objective in equation 3a with an upper bound that we can also minimise with the same methods. This model is the most flexible choice of prior, however, it is more expensive to fit than the realNVP or the VAMP prior due to the additional level of hierarchy and the resulting bounding and inference step.

## 3 EXPERIMENTS

**Datasets** We evaluate our models on the following datasets. Swiss roll and Coil20 are classic datasets for manifold learning. The Human Motion Capture dataset includes both periodic motion (walking

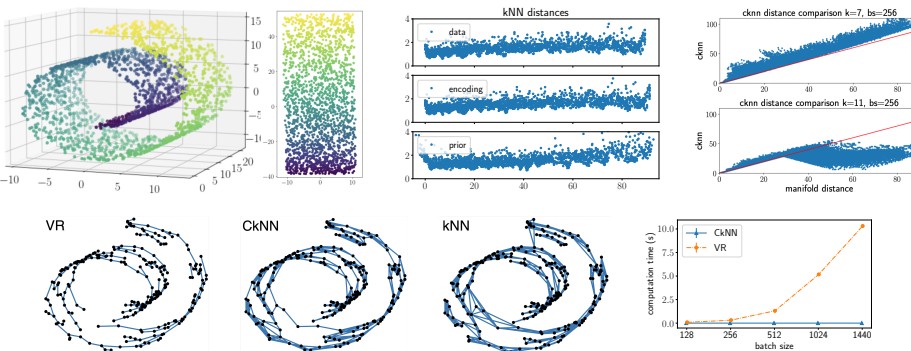

Figure 1: Top row: (left) the Swiss roll data (3d) with encoding result (2d) from CkNN-NVP, (middle) the kNN distance for the data and the encoding and prior samples resulting from a CkNN-NVP model, the distances are plotted w.r.t. the distance along the main axis on the manifold (right) illustrating a bridging vs correct graph construction by plotting the shortest path on the manifold vs in the CkNN graph. Bottom row: graphs construction examples for CkNN with $k = 9$ and $\delta = 0.9$, kNN with $k = 4$. (Bottom-right) computation time of $L_{\text{topo}}$ with the distance matrix as input and without back-propagation on batches from Coil20 ($32 \times 32$ pixels). We show the mean over 20 runs.

and jogging) and line motion (balancing), from which we can easily observe and identify the topology of the data. Cifar10 (in the Appendix) is another type of dataset that can be used to evaluate our models in the general case, not limited to known manifolds.

**Models** The AE-VR and VAE-VR are the models from (Moor et al., 2020), and VAE-SNE from (Graving & Couzin, 2020), trained with methods in Section 2.2. For a comprehensive legend of model labels please check Table 6 in the Appendix.

**Evaluation metrics** To evaluate our methods, we compute standard metrics on Swiss roll, CMU human motion, and Coil datasets. We use four metrics from (Moor et al., 2020) to evaluate the models, i.e., $\text{MRRE}_{z \to x}$, $\text{MRRE}_{x \to z}$, trustworthiness and continuity that are defined as follows. (i) $\text{MRRE}_{x \to z}$ (Moor et al., 2020) measures the changes between distance rankings as the data is encoded. The baseline ranking is computed w.r.t. the kNN graph ($k = 9$) in the data space. (ii) $\text{MRRE}_{z \to x}$ (Lee & Verleysen, 2009) is the same measure but with the baseline ranking computed w.r.t. the kNN graph of the encodings. trustworthiness Venna & Kaski (2006) evaluates the preservation the $k$ nearest neighbours during encoding while (iv) continuity (Venna & Kaski, 2006) evaluates it for the decoding. Note that all measures are based exclusively on the $k$ nearest neighbours and thus might disadvantage somewhat SNE and UMAP. We choose $k = 9$ for all experiments. Additionally, since we have the ground truth of the Swiss roll dataset, we compute the linear correlation between the shortest path on the data manifold and Euclidean distance on the latent space. For the Coil20 dataset, the neighbours of an image of an object is given by the camera angles, we compute the linear correlation between the input data and the latent encodings based on this neighbourhood during evaluation. Standard deviations on metrics are computed on 50 MC samples from $q_\phi(z|x)$.

Table 1: $\text{MRRE}_{z \to x}$ on MOCAP, smaller better.

|  | AE | VAE | NVP | VHP | VAMP |
|---|---|---|---|---|---|
| CkNN | **0.005** | 0.011(0.000) | 0.013(0.000) | **0.008(0.000)** | 0.013(0.000) |
| VR | 0.007 | 0.017(0.000) | 0.018(0.000) | 0.018(0.000) | 0.012(0.000) |
| SNE | 0.007 | 0.031(0.001) | 0.027(0.000) | 0.025(0.000) | 0.028(0.000) |
| UMAP | 0.008 | - | - | - | - |

Table 2: $\text{MRRE}_{x \to z}$ on MOCAP, smaller better.

|  | AE | VAE | NVP | VHP | VAMP |
|---|---|---|---|---|---|
| CkNN | **0.004** | 0.009(0.000) | 0.011(0.000) | **0.006(0.000)** | 0.012(0.000) |
| VR | **0.004** | 0.014(0.000) | 0.015(0.000) | 0.016(0.000) | 0.009(0.000) |
| SNE | 0.005 | 0.046(0.001) | 0.032(0.001) | 0.027(0.000) | 0.030(0.000) |
| UMAP | 0.005 | - | - | - | - |

Table 3: continuity on MOCAP, larger better.

|  | AE | VAE | NVP | VHP | VAMP |
|---|---|---|---|---|---|
| CkNN | **0.997** | 0.992(0.000) | 0.990(0.000) | **0.995(0.000)** | 0.989(0.000) |
| VR | 0.996 | 0.987(0.000) | 0.986(0.000) | 0.985(0.000) | 0.992(0.000) |
| SNE | 0.996 | 0.954(0.001) | 0.969(0.001) | 0.974(0.000) | 0.971(0.000) |
| UMAP | 0.996 | - | - | - | - |

Table 4: trustworthiness on MOCAP, larger better.

|  | AE | VAE | NVP | VHP | VAMP |
|---|---|---|---|---|---|
| CkNN | **0.995** | 0.990(0.000) | 0.988(0.000) | **0.993(0.000)** | 0.988(0.000) |
| VR | 0.993 | 0.984(0.000) | 0.984(0.000) | 0.983(0.000) | 0.990(0.000) |
| SNE | 0.994 | 0.965(0.001) | 0.974(0.000) | 0.976(0.000) | 0.972(0.000) |
| UMAP | 0.993 | - | - | - | - |

Table 5: Results on Coil20

| | AE-CkNN | AE-VR | AE-SNE | VAE-CkNN | NVP-CkNN | VHP-CkNN | VAMP-CkNN | VAE-VR | NVP-VR | VHP-VR | VAMP-VR |
|---|---|---|---|---|---|---|---|---|---|---|---|
| $MRRE_{z \to x}$ | 0.010 | 0.048 | **0.009** | 0.031(0.000) | **0.017(0.000)** | 0.034(0.000) | 0.028(0.000) | 0.023(0.000) | 0.034(0.000) | 0.036(0.000) | 0.035(0.000) |
| $MRRE_{x \to z}$ | **0.004** | 0.010 | 0.011 | 0.007(0.000) | **0.005(0.000)** | 0.007(0.000) | 0.008(0.000) | 0.008(0.000) | 0.008 (0.000) | 0.009(0.000) | 0.008(0.000) |
| continuity | **0.994** | 0.986 | 0.988 | 0.989(0.000) | **0.992(0.000)** | 0.989(0.000) | 0.987(0.000) | 0.991(0.000) | 0.991(0.000) | 0.989(0.000) | 0.990(0.000) |
| trustworthiness | 0.983 | 0.940 | **0.988** | 0.952(0.000) | **0.970(0.000)** | 0.954(0.000) | 0.955(0.000) | 0.968(0.000) | 0.950(0.000) | 0.950(0.000) | 0.950(0.000) |

**Hyperparameters** We consider as general hyper-parameters the batch size, encoder, decoder and prior architectures, the constraint bounds $\xi_{rec}$ and $\xi_{topo}$, the annealing rate $\eta$, and a switch variable whether to turn on the main objective only after the first constraint satisfaction occurred. Furthermore, for CkNN we consider as hyper-parameter the length scale $\delta$, and the number of the neighbours $k$ while for t-SNE the length scale. We use the ADAM optimiser (Kingma & Ba, 2015) with learning rate 0.001 as implemented in PyTorch (Paszke et al., 2019). For each dataset, we use the same encoder, decoder and prior architectures across on all methods. In the Swiss roll latent space experiment, VR-AE requires larger batch size than the VR-VAE-based and CkNN-based models.

**Illustrative example: Swiss roll** The Swiss roll dataset e.g. (Pedregosa et al., 2011) is a standard artificial dataset used in non-linear dimensionality reduction and data visualisation which has several properties that can illustrate the benefits and pitfalls of various algorithms. The data is sampled as $(t, s) \sim \mathcal{U}_{[3\pi/2, 3\pi]} \times \mathcal{U}_{[0,21]}$ and transformed via $x(t, s) = (t \cos(t), s, t \sin(t))$. It has two properties that are particularly interesting to us: (i) the data is not uniformly distributed on the manifold defined by $[3\pi/2, 3\pi] \times [0, 21]$ because the density decreases with increasing $t$; and (ii) the periodic functions give rise to a folding with increasing radius thus confusing nearest neighbour methods when we do not use enough data. The latter manifest itself through the graph constructions choosing connections that "bridge" the manifold.

In the top-middle panel of Figure 1 we show the kNN distances for each data point and encoding (top and middle) as well as from the learned prior (bottom) when plotted against the main axis of the manifold and the main axis of the 2-d encoding, respectively. We can see that the encoding not only preserves the local distances but also, as a consequence, the density of the data along the main axis. The prior samples also show a similar pattern proving that latent space samples have similar kNN characteristics (see appendix A.2 for a visualisations of the learned prior). To detect bridging we can compare the shortest path in the graphs to the true shortest paths on the manifold, an example is shown in the top-right panel of Figure 1 for a low batch size. We compute the true shortest paths by solving numerically the boundary value problems resulting from the corresponding Euler-Lagrange equations. On this dataset CkNN and VR perform similarly well with CkNN performing slightly better. The rest of the methods are unable to recover correctly the 2D manifold. Due to the shape of the manifold they recover this might be due to the bridging effects/too many edges, see AE-SNE results in appendix A.2. As one can expect, the performance of the models is dependent on the chosen batch size. In Figure 5 in the Appendix we compare the auto-encoder model with VR and CkNN on the Swiss roll dataset using various batch sizes. The higher connectivity of CkNN seems to lead to better performance for smaller batch sizes.

**Human Motion Capture dataset** We use the Human Motion Capture dataset with 33 sequences of various lengths representing five movements, i.e., walking, jogging, punching, balancing and kicking (http://mocap.cs.cmu.edu/). The dataset includes 50-dimensional joint angles following the data-preprocessing in (Chen et al., 2015) and the data is scaled to the range $[0, 1]$. In total, there are 13355 configurations of joint angles which we consider as our dataset. From this dataset, we uniformly select 80 % for training and 20 % for testing. For the training data, we add a Gaussian noise of $\sigma = 0.03$. For this dataset, we use a batch size of 512, $k = 9$, and for each model, we varied as hyper-parameters $\xi_{rec}$, $\xi_{topo}$, $\delta$, and the annealing rate $\eta$.

Tables 1, 2, 3 and 4 are the results on human motion dataset. We sample the data from latent space 50 times and obtain the stochastic results in the tables. Generally, we can observe the following. (i) Learning generative models has typically lead to worse metrics than in AE models. We expect that this is due to the additional regulariser arising from learning the prior. (ii) Results for VAEs and hierarchical VAEs are comparable with clear advantages only in case of SNE. (iii) Generally, CkNN leads to improved metrics when compare to other graph construction methods and losses (SNE and UMAP). The latent representation and the learned priors are shown in the Appendix.

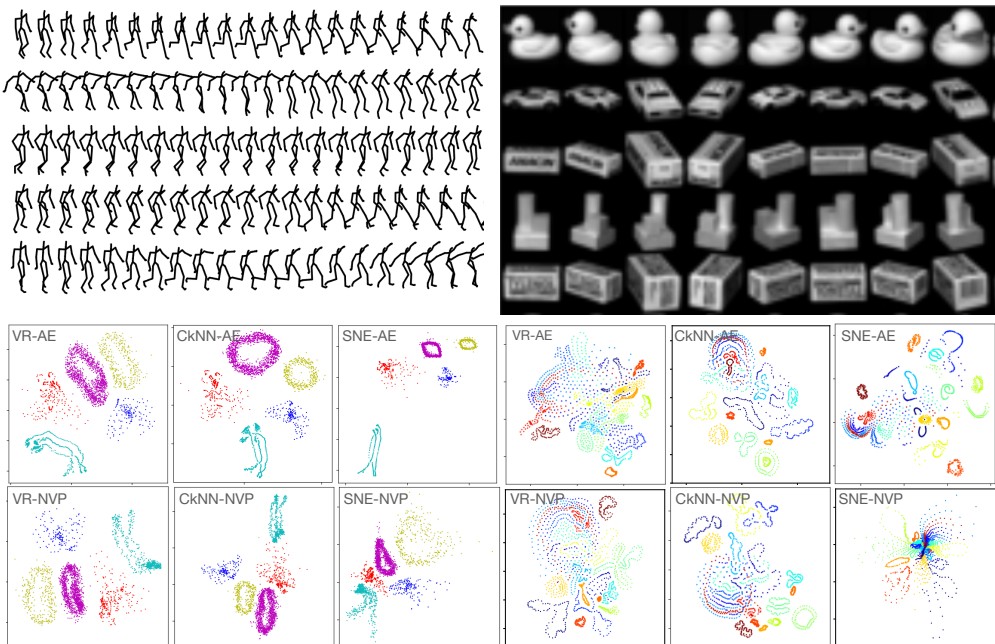

Figure 2: Real world datasets and 2d encodings. Top: the panels show examples form the Human Motion Capture (left) and the Coil-20 (right) datasets. Bottom: Two dimensional projections of samples from the datasets when computed using the shown methods. For the human motion dataset, the models preserves the topology – walking (magenta) and jogging (yellow) as circles and balancing (green) as lines. For the Coil20, some round objects have no obvious difference with different camera angles, so that they are distributed in small regions. Cars and cuboids are similar to each other, and distribute next to each other. Note that topological consistency for periodic motions/structures requires closed non-intersecting loops, not necessarily circles.

**Coil-20 dataset** We use Columbia Object Image Library with 20 objects (Coil-20) (Nene et al., 1996). We sample the data from latent space 50 times and obtain the stochastic results in the table. The dataset consists of 72 images of each object that were taken for by uniformly rotating the camera around the object. This results 1440 images in total. The backgrounds were pre-processed to be black, and the images were cropped to $32 \times 32$ pixels with grey scale. Since the dataset is small, we use the maximum possible batch size of 1440, $k = 9$, and for each model, we varied $\xi_{\text{rec}}$, $\xi_{\text{topo}}$, $\delta$, and the annealing rate $\eta$.

In Table 5 we show the results on Coil-20. We can observe that NVP-CkNN performs generally the best and from the AE models. The round objects which have small amount of pixels changed with different camera angles distribute quite small in the latent space of CkNN-AE. However, more than half of the objects in SNE-AE latent space have no obvious size difference. Some objects (e.g., cuboids) are two circles in the latent space, since they have similar images between the back and front sides. It is reasonable that an object locates into a big circle even though they are not similar, since the space there is large enough.

## 4  RELATED WORK

**Manifold learning** Manifold learning encompassed a large variety of dimensionality reduction methods designed with a different guiding principle compared deep generative models that use an auto-encoding view of Bayesian inference. The methods in manifold learning generally search for neighbourhood structures or define the affinities between points and aim to embedd high-dimensional data into a low-dimensional space while conserving some of these properties/affinities. Methods include, e.g., Locally Linear Embedding (LLE) (Roweis & Saul, 2000), Local Tangent Space Alignment (LTSA) (Zhang & Zha, 2004), Multi-dimentional Scaling (MDS) (Carroll & Arabie, 1998), t-distributed Stochastic Neighbour Embedding (tSNE) (van der Maaten & Hinton, 2008), Uniform Manifold Approximation and Projection (UMAP) (McInnes et al., 2018), and Isometric

Mapping (ISOMAP) (Tenenbaum et al., 2000). Deep generative models generally do not explicitly encode neighbourhood information, however, such properties often emerge. Several follow-up studies combine VAEs and manifold learning. Topo-AE (Moor et al., 2020) and Connectivity-Optimized Representation Learning (Hofer et al., 2019) construct graphs using persistent homology, which preserves the topology between data and latent spaces. In (Schönenberger et al., 2020) a different graph construction method based on witness complexes (Silva & Carlsson, 2004) is presented. Their model uses an additional dataset witness dataset for graph construction and non-identical graph construction methods in the data and latent space, respectively Based on Topo-AE, (Li et al., 2021) define local distance preserving invertible encoder-decoder models, VAE-SNE (Graving & Couzin, 2020) optimises pairwise similarity between the distributions of the data and latent spaces to preserve the local neighbourhood. Parameterised UMAP (Sainburg et al., 2021) adapts the original UMAP algorithm (projection only) to an AE framework by using neural networks. Neighbourhood Reconstructing Auto-encoder (NRAE) (Lee et al., 2021) proposed to preserve the neighbourhood structure by minimising the distances between the output of a data point on the gradient direction of the decoder and its neighbours.

**Constraint optimisation for VAEs** VAE models are typically challenging to train due to the difficulty of balancing the reconstruction and compression (KL) term during training (Sønderby et al., 2016; Alemi et al., 2018). Several non-adaptive annealing schedules have been proposed to slowly turn on the KL-terms during training e.g. (Sønderby et al., 2016) or that anneal according to a task-specific utility function (Higgins et al., 2017). Taming VAE (Rezende & Viola, 2018) propose to use an annealing scheme derived from a constrained optimisation approach. VaHiPrior (Klushyn et al., 2019) adapted (Rezende & Viola, 2018) to (two level) hierarchical generative models. In (Zhao et al., 2018) the authors study the constrained optimisation approach in several of VAEs and GAN models establishing formal similarities between ELBO/GAN losses with information theoretic regularisers/constraints (adapting/fixing the Lagrange multipliers depends on the connection they aim to establish). In our work, we propose a simple (fully fitted) constrained optimisation with the reconstruction and topological losses as constraints. In our experience, this approach significantly improved training performance compared to various combinations of regularisers (fixed multipliers) and KL annealing schedules.

**Learning priors for VAEs** A Gaussian prior for VAEs can often lead to over-regularization. Therefore, various flexible priors were developed. In (Dilokthanakul et al., 2016) a Gaussian mixture is proposed as the prior. Variational Mixture of Posteriors prior (VampPrior) Tomczak & Welling (2018) learns a prior that is defined based on the optimal empirical Bayes prior using learned pseudo-data. Klushyn et al. (2019) recast learning the prior as learning an equivalent (two level) hierarchical VAE model. We implement a choice of priors with various degree of complexity representing a range of flexibility vs computational complexity trade-offs.

## 5 CONCLUSION AND FUTURE WORK

In this paper we propose the CkNN graph construction method for local distance preserving auto-encoder and hierarchical variational auto-encoder models. The CkNN graph is not only inexpensive to compute but it also leads to comparable results when compared to the persistent homology (VR), SNE, and UMAP as shown in Section 3. The additional hyper-parameters $k$ and $\delta$ that CKNN requires are typically easy to tune and hence CkNN can represent a viable alternative option. To improve training and to achieve a good balance between different loss terms, we use a constraint optimisation framework that results in hyper-parameters (constraint bounds) that are easier to tune than weight parameters in a regularisation setting. Based on the experiments presented in Section 3, we conclude that CkNN-AE performs on average better than other AE-based models. CkNN performs well in models with no clear separation of clusters/no clusters in data (e.g. Swiss-roll). CkNN based generative models have an overall good performance among generative models. Flexible priors significantly improve the results of SNE-based models, while they seem to have lesser impact on the metrics in other models. Additionally, the CkNN computation is typically faster than VR, especially for large batch sizes even if it comes with two hyper-parameters that, as we experienced, are easy to fit.

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

# A  APPENDIX

## A.1  IMPLEMENTATION DETAILS

**Details of the baseline models**  The graph in AE-UMAP is built only in the data space and uses the whole training dataset. In this method the batch size refers to the number of pairs while other methods are the number of data points. Consequently, the batch size has small influence for the training. On the contrary, in other models the graph are built both in data- and latent- space; using a batch of data points. We use the same batch size for all methods. In Topo-AE in Moor et al. (2020), which we denote by (VR), we use the graph construction and topological loss computation code from `https://github.com/BorgwardtLab/topological-autoencoders`. In Moor et al. (2020), the topological regulariser is computed by summing the loss over the batch data. To avoid the balance between the reconstruction and the regulariser to be influenced by the batch size, we compute the loss by computing the mean over the graph edges. This results in having more informed choices for the constraint bound. The VR-based models count the intersection graphs twice as defined in Moor et al. (2020), while our CkNN-based models count only once according to Equation (1). This assigns a slightly higher weight to the symmetric difference of the graphs and has no negative effect on training. In the evaluation, except Cifar10, we use the whole test dataset as a batch to compute the metrics. In the tables, **bold** and **bold** indicates the best results in the AE-based and VAE-based models, respectively. The hyper-parameter search is carried out on a Nvidia DGX (Tesla V100 GPUs, 8 Gbs for each GPU) using Polyaxon 0.6.1 [1]. Our code is implemented using PyTorch 1.9.1. Each experiment uses one GPU.

In Table 6, we show references of the models. All of these models are trained with the methods presented in Section 2.2 except (Sainburg et al., 2021). They are identical from the original papers.

Table 6: Models.

|      | AE                    | VAE                      | NVP         | VHP         |
|------|-----------------------|--------------------------|-------------|-------------|
| N/A  | -                     | Kingma & Welling (2014)  | -           | -           |
| CkNN | this paper            | this paper               |             |             |
| VR   | Moor et al. (2020)*   | Moor et al. (2020)*      | this paper  | this paper  |
| SNE  | this paper            | Graving & Couzin (2020)* | this paper  | this paper  |
| UMAP | Sainburg et al. (2021)| -                        | -           | -           |

## A.2  SWISS ROLL ON LATENT SPACE

**Computing shortest path on the manifold**  Since for this artificial dataset the manifold is defined by $(x(t), y(s), z(t)) = (t\cos(t), s, t\sin(t))$, $(t, s) \sim \mathcal{U}_{[3\pi/2, 3\pi]} \times \mathcal{U}_{[0,21]}$, we can compute the shortest path between two points $(t_0, s_0)$ and $(t_1, s_1)$ by finding the functions $(t(u), s(u))$, $u \in [0, 1]$ that minimise the functional

$$F(s, t) = \int_0^1 du \left[ \dot{x}(t(u))^2 + \dot{y}(s(u))^2 + \dot{z}(t(u))^2 \right]^{1/2}.$$

By taking the variation of $F$ w.r.t. $s(u)$ and $t(u)$ and setting them to 0, we obtain two independent differential equations $\ddot{s}(u) = 0$ and $t(u)\dot{t}(u)^2 + (1 + t(u))\ddot{t}(u) = 0$. The optimal function $s(u)$ is linear passing through $s_0$ and $s_1$. We did not find an analytic solution to the equation for $t(u)$ and thus we solved the corresponding boundary value problem numerically (with boundaries $t_0$ and $t_1$).

**Dataset**  We randomly generate $2^{16}$ training data by sampling $t$ and $s$. For testing we sample 2048 points, and for the distance comparison we sample 1024 points. We use a batch size $n_{\text{batch}} = 256$.

**Latent space**  As shown in Figure 3, AE-CkNN achieves the geometrically most faithful projection of the original manifold across all AE-based models. In generative models, CkNN and VR show a consistently faithful projection of both the shape and density of the original manifold. VR-AE models do not achieve a good projection but increasing the batch size can improve the performance. We trained VR-AE with a batch size 512, and we have achieved similar results like the best performing models (see Figure 5). Using VAEs with learned priors seems to improve the SNE models when compared to VAE-SNE, while the learning the prior has less significant effect on other models.

---

[1]`https://polyaxon.com`

Table 7: $\text{MRRE}_{z \rightarrow x}$ on Swiss roll, smaller better.

|  | AE | VAE | NVP | VHP | VAMP |
|---|---|---|---|---|---|
| CkNN | **0.000** | **0.000(0.000)** | **0.000(0.000)** | **0.000(0.000)** | **0.000(0.000)** |
| VR | **0.000** | **0.000(0.000)** | **0.000(0.000)** | **0.000(0.000)** | **0.000(0.000)** |
| SNE | **0.000** | 0.036(0.001) | 0.035(0.001) | 0.035(0.001) | 0.035(0.001) |
| UMAP | **0.000** | - | - | - | - |

Table 8: $\text{MRRE}_{x \rightarrow z}$ on Swiss roll, smaller better.

|  | AE | VAE | NVP | VHP | VAMP |
|---|---|---|---|---|---|
| | **0.000** | **0.000(0.000)** | **0.000(0.000)** | **0.000(0.000)** | **0.000(0.000)** |
| | **0.000** | **0.000(0.000)** | **0.000(0.000)** | **0.000(0.000)** | **0.000(0.000)** |
| | **0.000** | 0.039(0.001) | 0.041(0.001) | 0.051(0.001) | 0.034(0.001) |
| | **0.000** | - | - | - | - |

Table 9: continuity on Swiss roll, larger better.

|  | AE | VAE | NVP | VHP | VAMP |
|---|---|---|---|---|---|
| CkNN | **1.000** | **1.000(0.000)** | **1.000(0.000)** | **1.000(0.000)** | **1.000(0.000)** |
| VR | **1.000** | **1.000(0.000)** | **1.000(0.000)** | **1.000(0.000)** | **1.000(0.000)** |
| SNE | **1.000** | 0.963(0.001) | 0.961(0.001) | 0.950(0.001) | 0.968(0.001) |
| UMAP | **1.000** | - | - | - | - |

Table 10: trustworthiness on Swiss roll, larger better.

|  | AE | VAE | NVP | VHP | VAMP |
|---|---|---|---|---|---|
| | **1.000** | **1.000(0.000)** | **1.000(0.000)** | **1.000(0.000)** | **1.000(0.000)** |
| | **1.000** | **1.000(0.000)** | **1.000(0.000)** | **1.000(0.000)** | **1.000(0.000)** |
| | **1.000** | 0.965(0.001) | 0.966(0.001) | 0.965(0.001) | 0.967(0.001) |
| | **1.000** | - | - | - | - |

Table 11: Linear correlation on Swiss roll. Pearsonr linear correlation between the distance in the latent space and distance on the data manifold. 1 or -1 are completely correlated, 0 is not correlated. It is corresponding to Figure 6.

|  | AE | VAE | NVP | VHP | VAMP |
|---|---|---|---|---|---|
| CkNN | **1.00** | **1.00(0.00)** | **1.00(0.00)** | **1.00(0.00)** | **1.00(0.00)** |
| VR | 0.70 | **1.00(0.00)** | **1.00(0.00)** | **1.00(0.00)** | **1.00(0.00)** |
| SNE | 0.32 | 0.46(0.01) | 0.44(0.01) | 0.72(0.01) | 0.85(0.00) |
| UMAP | 0.99 | - | - | - | - |

AE-UMAP does reasonably well in recovering the shape but not the density—the width of the high density region (dark blue) distributes larger than the low density area (yellow). AE-UMAP and SNE-based models have no distance preserving, therefore, it is reasonable that they do not outperform other regularisers. In addition, two failure cases are shown in Figure 7.

**Priors** Figure 4 shows that the learned priors match well the corresponding latent spaces (Figure 3) both in shape and density. For example, the upper (dark blue) region of Figure 3g has a higher density than the lower (yellow). A similar pattern is present in the data generated from the prior as shown in Figure 4a. VHP and VAMP match the shape of the encodings better than NVP in the SNE experiments. Without turning off the KL before first satisfied, the VAMP prior is difficult to be learned, but VHP and NVP are not affected by this option.

**Distance in the latent space** Figure 6 shows the correlation between the Euclidean distances in the latent space and the distances on the manifold. A straight line in the figure shows that the two distances are correlated. AE-CkNN and VAE-based CkNN and VR models show the best results. AE-UMAP is slightly worse than the best models.

**Metrics** Table 7, 8, 9, and 10 show the standard metrics on the Swiss roll dataset, and Table 11 shows the numerical results of the linear correlation in Figure 6. We sample the data from latent space 50 times and obtain the stochastic results in the tables. CkNNs achieve the best results from all AE-based models, while VR and CkNN are the best in the VAE-based models. The distance correlation has been significantly improved by the learning the priors, although other four metrics of the learned prior for the SNE have similar results as the VAE-SNE.

### A.3 LATENT SPACES AND PRIORS ON HUMAN MOTION DATASET

Figures 9 and 8 show the latent spaces and priors of the human motion experiments of Section 3. We observe similar results like in the Swiss roll dataset. The fixed prior in VAE-SNE over-regularises the posteriors, which results in, for instance, the jogging motion being hardly a periodic in the latent space. On the contrary, SNE with learned priors shows clear periodic patterns for walking and jogging, and lines for balancing. The priors have similar shapes and density like the posteriors—the walking density has a higher density than the jogging, since we have more walking samples in the dataset.

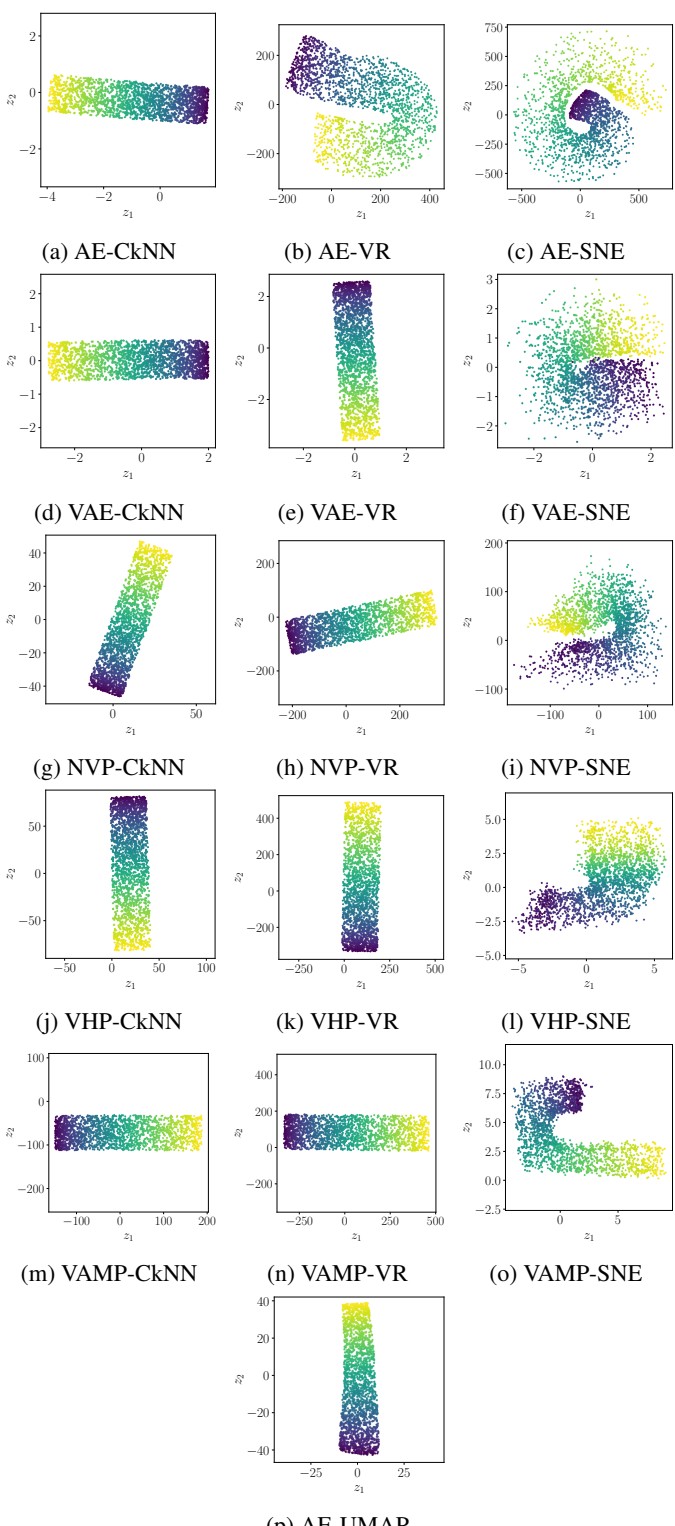

Figure 3: Latent space of Swiss roll. The batch size is 256. The color represents $t$. A rectangle in the latent space indicates that the model has learned the intrinsic properties of the Swiss roll. See more details in Section A.2.

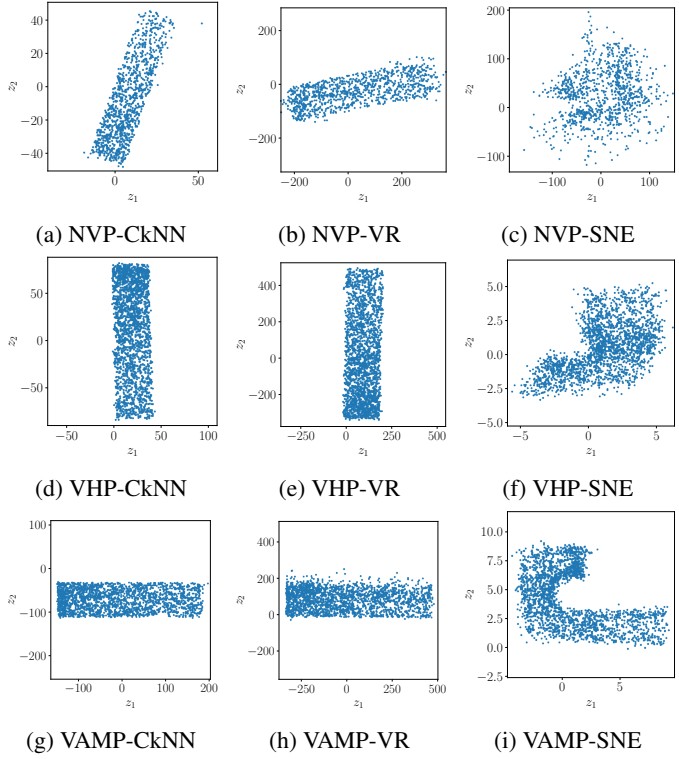

Figure 4: Learned priors of Swiss roll dataset. The corresponding posteriors are shown in Figure 3. See more details in Section A.2.

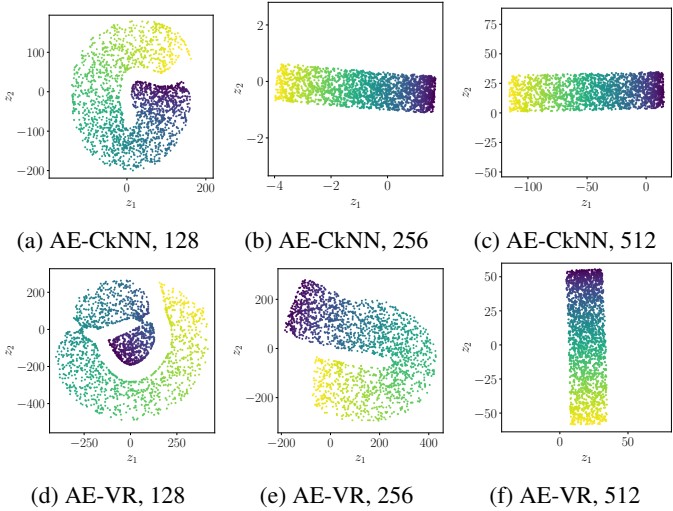

Figure 5: Latent space of Swiss roll with different batch size. Compared with the AE-CkNN, the AE-VR requires larger batch size to learn the intrinsic properties of the Swiss roll.

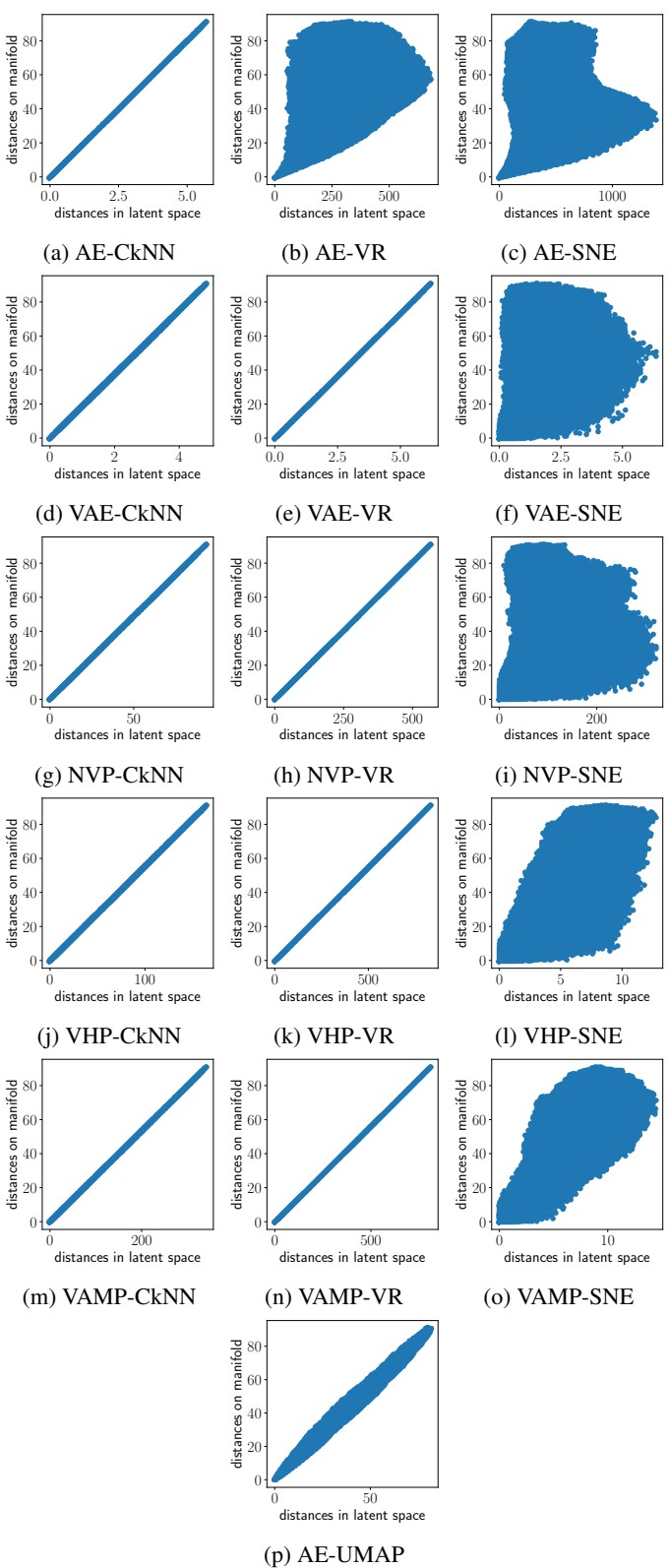

Figure 6: Comparison of the Swiss roll distances. Linear correlation between the distances indicates that the Euclidean distance in latent space is able to represent the distances on the data manifold. The numerical results are shown in Table 11. See more details in Section A.2.

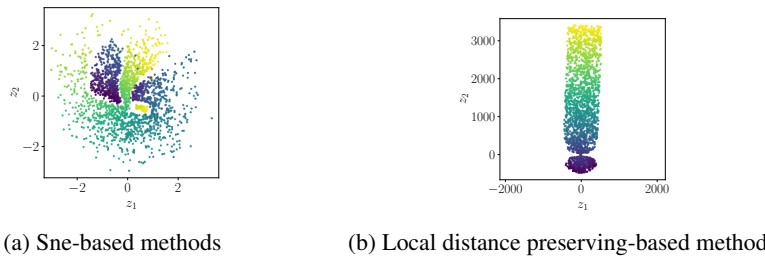

(a) Sne-based methods        (b) Local distance preserving-based methods

Figure 7: The latent space of failure cases of Swiss roll dataset. The SNE losses between twisted and untwisted latent spaces have no obvious difference. However, it has obvious difference in the local distance preserving loss. Therefore, even with the cases that the latent representation is difficult to be untwisted using local distance preserving, we can easily select the well trained model. See more details in Section A.2.

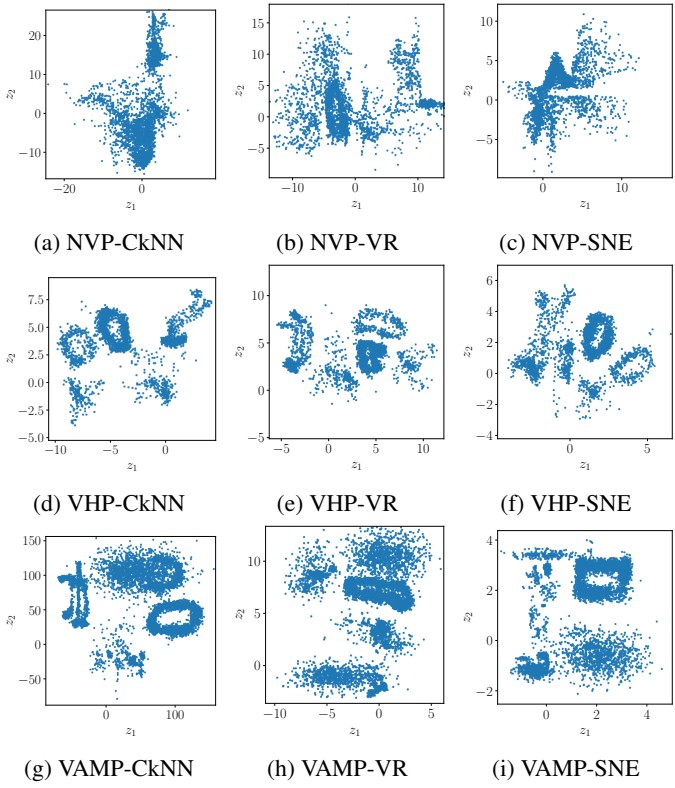

(a) NVP-CkNN      (b) NVP-VR      (c) NVP-SNE

(d) VHP-CkNN      (e) VHP-VR      (f) VHP-SNE

(g) VAMP-CkNN      (h) VAMP-VR      (i) VAMP-SNE

Figure 8: Learned priors of human motion dataset. The corresponding posteriors are shown in Figure 9. See more details in Section A.3.

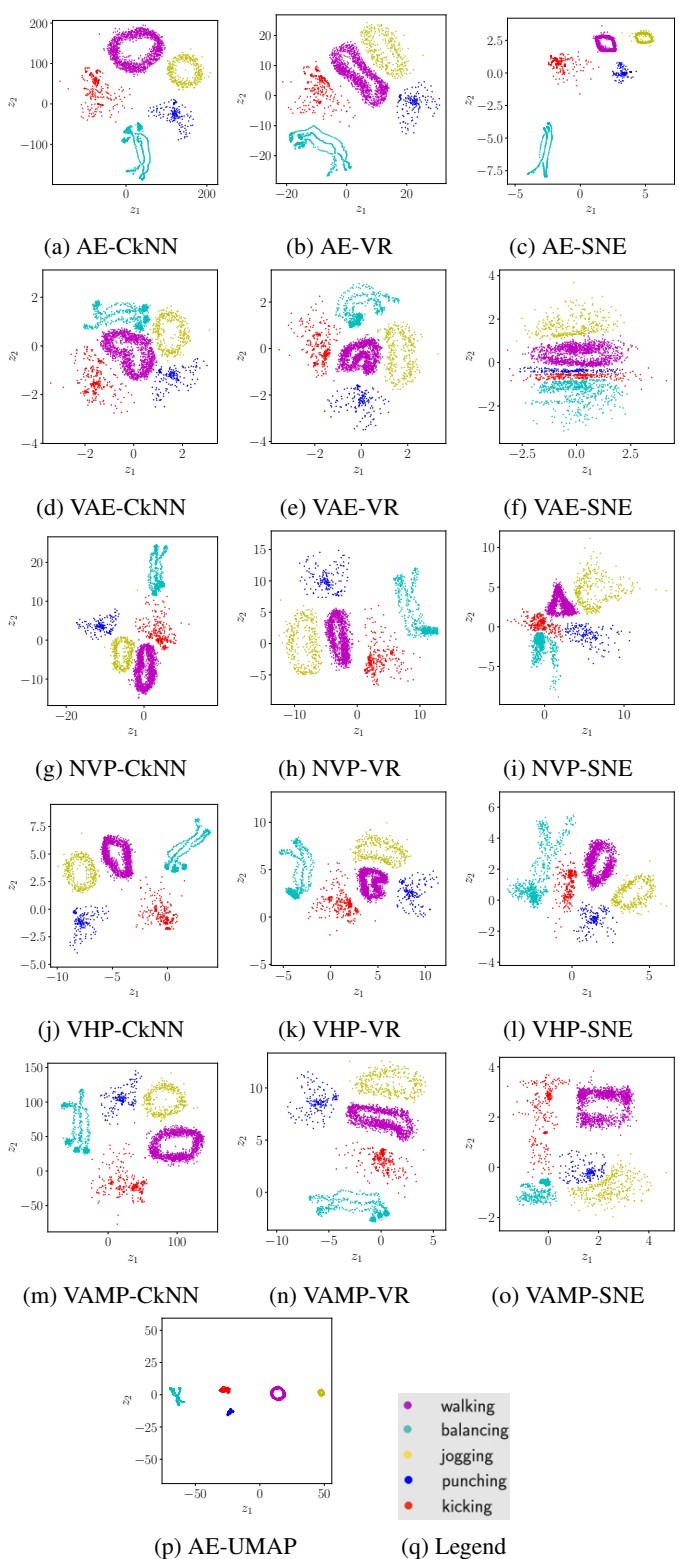

Figure 9: Latent space of human motion dataset. Except VAE-SNE, other models have clear boarders of the five classes and preserve the topology, i.e., walking and jogging as periodic and balancing as line latent representation. See more details in Section A.3.

## A.4 RESULTS OF COIL20

In this section, we show additional results from the Coil20 experiments (Section 3). Figure 12 shows examples of the data reconstruction of the models. Figure 10 shows the latent spaces corresponding to the results shown in Table 5. The learned priors (Figure 11) have similar distribution like the encodings/posteriors.

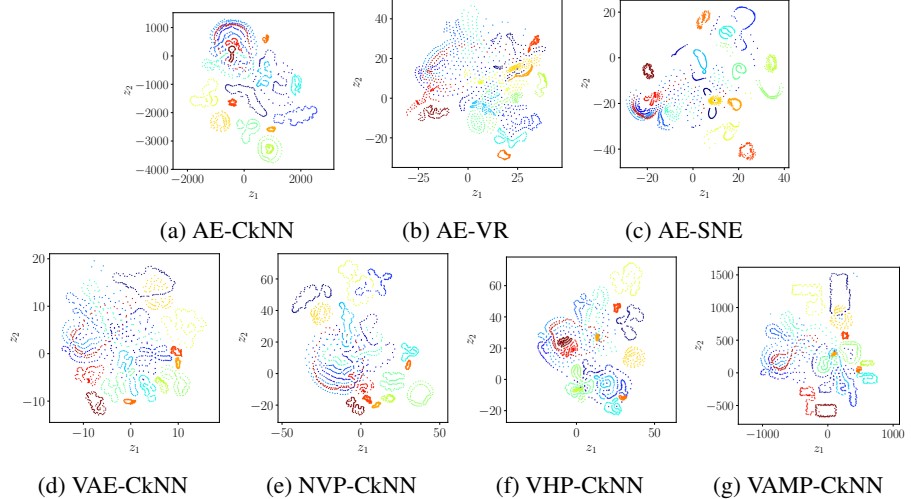

Figure 10: Latent space of Coil20 dataset. See more details in Section A.4.

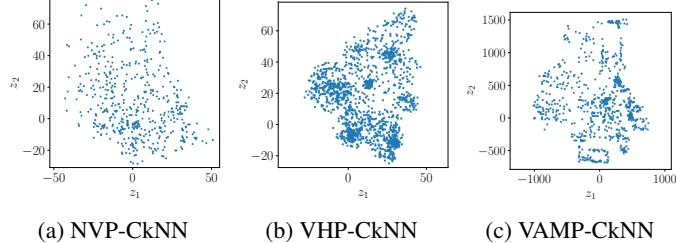

Figure 11: Learned priors of Coil20 dataset. See the posteriors in Figure 10. See more details in Section A.4.

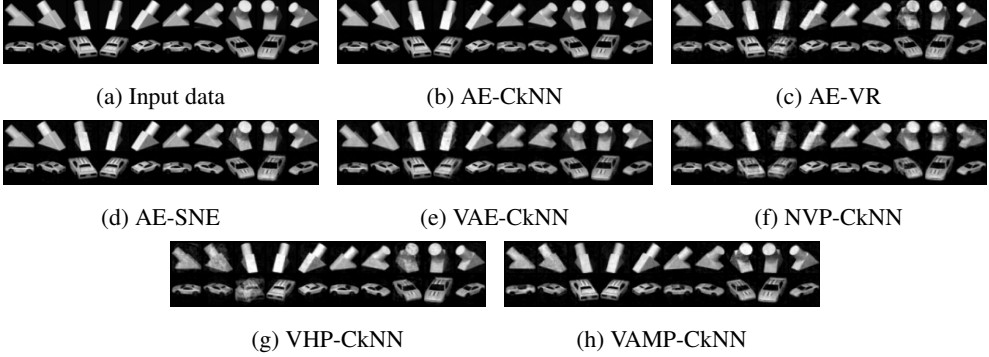

Figure 12: Input data and the reconstruction of Coil20. See more details in Section A.4.

## A.5 EXPERIMENTS ON CIFAR10

Table 12: AE-based models on Cifar10. The results are mean (STD).

|  | AE | AE-CkNN | AE-VR | AE-SNE |
|---|---|---|---|---|
| l_MRRE_ZX | 0.010(0.001) | **0.001(0.000)** | **0.001(0.000)** | 0.007(0.000) |
| l_MRRE_XZ | 0.011(0.000) | **0.001(0.000)** | **0.001(0.000)** | 0.007(0.000) |
| l_continuity | 0.988(0.001) | **1.000(0.000)** | **1.000(0.000)** | 0.993(0.000) |
| l_trustworthiness | 0.989(0.001) | **1.000(0.000)** | **1.000(0.000)** | 0.993(0.000) |

Table 13: VAE-based models on Cifar10. The results are mean (STD).

|  | VAE | VAE-CkNN | NVP-CkNN | VHP-CkNN | VAMP-CkNN | VAE-VR | NVP-VR | VHP-VR | VAMP-VR |
|---|---|---|---|---|---|---|---|---|---|
| l_MRRE_ZX | 0.128(0.006) | 0.002(0.000) | 0.002(0.000) | 0.002(0.000) | **0.001(0.000)** | 0.002(0.000) | 0.002(0.000) | 0.002(0.000) | 0.002(0.000) |
| l_MRRE_XZ | 0.082(0.004) | 0.002(0.000) | 0.002(0.000) | 0.002(0.000) | **0.001(0.000)** | 0.002(0.000) | **0.001(0.000)** | 0.002(0.000) | 0.002(0.000) |
| l_continuity | 0.907(0.004) | **0.999(0.000)** | **0.999(0.000)** | **0.999(0.000)** | **0.999(0.000)** | **0.999(0.000)** | **0.999(0.000)** | **0.999(0.000)** | **0.999(0.000)** |
| l_trustworthiness | 0.859(0.007) | **0.999(0.000)** | **0.999(0.000)** | **0.999(0.000)** | **0.999(0.000)** | **0.999(0.000)** | **0.999(0.000)** | **0.999(0.000)** | **0.999(0.000)** |

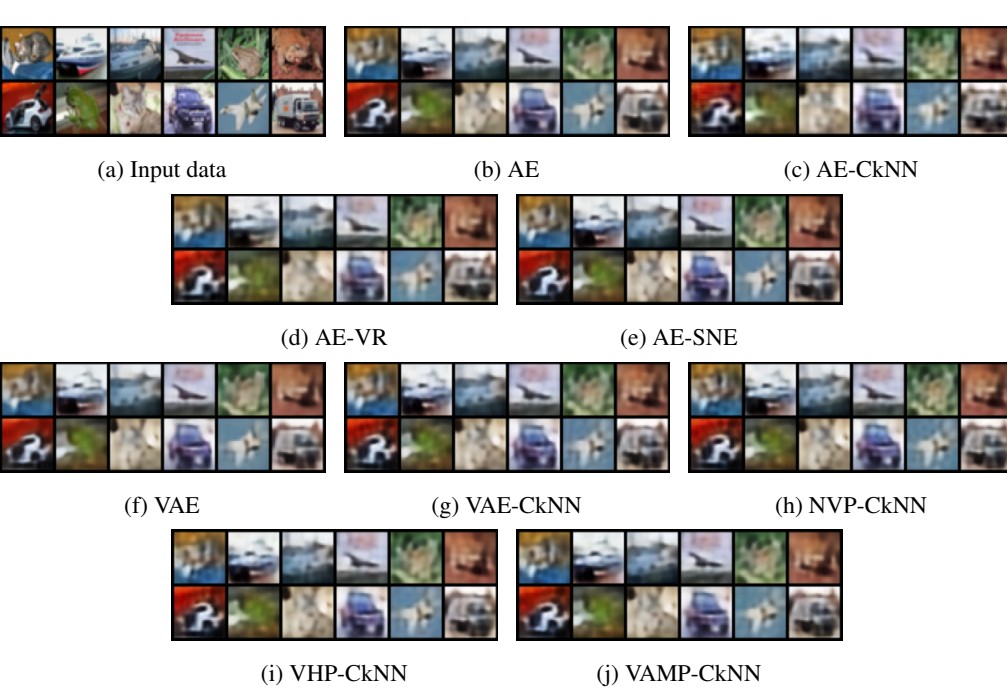

(a) Input data   (b) AE   (c) AE-CkNN

(d) AE-VR   (e) AE-SNE

(f) VAE   (g) VAE-CkNN   (h) NVP-CkNN

(i) VHP-CkNN   (j) VAMP-CkNN

Figure 13: Input data and the reconstructions of Cifar10 with 128 latent dimensions. The reconstruction is slightly blurry, but we can recognise the objects in the images. Increasing the latent dimensions or using alternative encoder/generator architectures could reconstruct more accurately, but it is not the focus of this paper. See more details in Section A.5.

Cifar10 is a standard dataset for classification with 10 classes and 6000 images per class with size $32 \times 32 \times 3$ (Krizhevsky et al., 2009). Here we only consider the image data ignoring the labels. We randomly select 80 % as the training data and the rest is the test data. The encoder and decoder architectures were taken from (https://uvadlc-notebooks.readthedocs.io) For this dataset, we use batch size 512, latent dimension 128, $k = 9$, and for each model, we varied $\xi_{\text{rec}}$, $\xi_{\text{topo}}$, $\delta$, and the annealing rate $\eta$.

All models have reasonably good reconstructions (see Figure 13). We use batch size 1000 for evaluation, and obtain the mean and STD of the metrics over batches (see Table 12 and 13). The AEs with manifold constraint have significant better metrics than the vanilla AE, although the AE-SNE yields to the local distance preserved models. Similarly, VAE is not comparable to models with a CkNN regulariser.

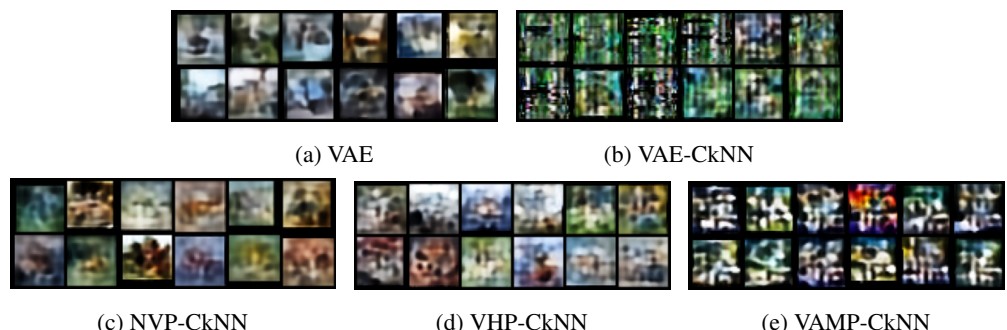

(a) VAE                                    (b) VAE-CkNN

(c) NVP-CkNN                    (d) VHP-CkNN                    (e) VAMP-CkNN

Figure 14: Samples from the prior of Cifar10. See more details in Section A.5.

Furthermore, we generate image samples from the priors using the generative models. Due to the topological regulariser, the posterior does not fit perfectly with the fixed prior (VAE-CkNN); therefore, it is challenging to sample meaningful images from the prior (see Figure 14b). However, NVP-CkNN and VHP-CkNN are able to generate images (e.g., dog) which are not in the training dataset.

### A.6 GRAPHS

In this section we show a few illustrative examples about graphs construction. In Figure 15 we show a simple dataset with a mixture of two uniform distributions; the artificial dataset definition is from (Berry & Sauer, 2019). The mixture distribution results in the two box-shapes having different densities. This gives rise to a bridging effect mentions in Section 2 of the main text; in standard kNN based approaches low density regions tend to connect to higher density ones due to the lack of close enough similar neighbours. VR constructs a spanning tree thus, as expected, there is only one bridging edge. Since CkNN uses local kNN distances for connectivity it helps to alleviate bridging. Another important property of graph construction is the resulting number of connected components. This can be relevant when we want to compute a shortest path interpolation in the latent space for visualisation purposes (see (Klushyn et al., 2019)). It is often difficult to balance having good enough connectivity and avoid bridging effects. In our experience CkNN often manages to find the right balance or is the easiest to tune w.r.t. $k$ and $\delta$. The results on the boxes and Swiss roll datasets are shown in Figure 16.

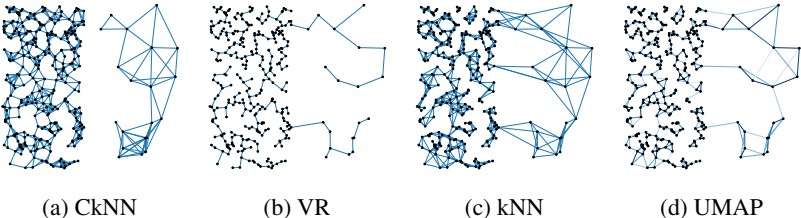

(a) CkNN      (b) VR      (c) kNN      (d) UMAP

Figure 15: Graphs for the two boxes dataset with 320 data points. For kNN and UMAP graphs we chose $k = 4$, while for CkNN we chose $k = 10$ and $\delta = 0.9$. The CkNN represents the data structure more accurately and distinguishes the two densities better. Higher $k$ values in kNN and UMAP lead to even more bridging connections while less components lead to more disconnected graphs. See more details in Section A.6.

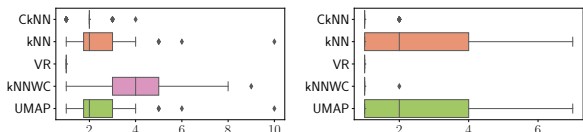

Figure 16: (left) The umber of components on the boxes dataset (see Figure 15) over 100 times with random of batch size 320. (right) The number of components of Swiss roll data over 100 times with random data with batch size 256. See more details in Section A.6.

A.7   ALGORITHM

---

**Algorithm 1** Training algorithm for the generative model (see Section 2).

---

**Hyper-parameters**: $n_{\text{batch}}, \xi_{\text{rec}}, \xi_{\text{topo}}, \eta_{\text{rec}}, \eta_{\text{topo}}, \text{InitialPhaseKL}$, topological hyper-parameters e.g. $\delta, k$

**Constants**: $\lambda_{\text{rec}}^0, \lambda_{\text{topo}}^0, \lambda_{\text{rec}}^{\max}, \lambda_{\text{topo}}^{\max}, \alpha$
Initialise $t = 0$
Initialise $\lambda_{\text{rec}} = \lambda_{\text{rec}}^0$
Initialise $\lambda_{\text{topo}} = \lambda_{\text{topo}}^0$
Initialise InitialPhaseRec = True
Initialise InitialPhaseTopo = True
Initialise InitialPhaseKL (True or False as HPs)
**while** training **do**
 Read current data batch $X_b$ of size $n_{\text{batch}}$
 Sample from variational posterior $Z_b \sim q_\phi(\cdot | X_b)$
 Build graphs from $X_b$ and $Z_b$, compute $L_{\text{rec}}(\theta, \phi; X_b)$ and $L_{\text{topo}}(\theta, \phi; X_b, Z_b)$
 Compute $c_{\text{rec}} = L_{\text{rec}} - \xi_{\text{rec}}$ (batch average)
 Compute $c_{\text{topo}} = L_{\text{topo}} - \xi_{\text{topo}}$ (batch average)
 $\hat{c}_{\text{rec}} \leftarrow (1 - \alpha)\,\hat{c}_{\text{rec}} + \alpha\,c_{\text{rec}}, (c_{\text{rec}}^{(0)} = c_{\text{rec}})$
 $\hat{c}_{\text{topo}} \leftarrow (1 - \alpha)\,\hat{c}_{\text{topo}} + \alpha\,c_{\text{rec}}, (c_{\text{topo}}^{(0)} = c_{\text{rec}})$
 **if** $c_{\text{rec}} < 0$ **and** InitialPhaseRec **then**
   InitialPhaseRec = False
 **end if**
 **if** $c_{\text{topo}} < 0$ **and** InitialPhaseTopo **then**
   InitialPhaseTopo = False
 **end if**
 **if** InitialPhaseKL **and** ¬InitialPhaseTopo **and** ¬InitialPhaseTopo **then**
   InitialPhaseKL = False
 **end if**
 **if** ¬InitialPhaseRec **then**
   $\lambda_{\text{rec}} \leftarrow \lambda_{\text{rec}} \cdot \exp\{\eta_{\text{rec}} \cdot \hat{c}_{\text{red}}\}$
   $\lambda_{\text{rec}} \leftarrow \text{clip}(\lambda_{\text{rec}}, \lambda_{\text{rec}}^{\max})$
 **end if**
 **if** ¬InitialPhaseTopo **then**
   $\lambda_{\text{topo}} \leftarrow \lambda_{\text{topo}} \cdot \exp\{\eta_{\text{topo}} \cdot \hat{c}_{\text{topo}}\}$
   $\lambda_{\text{topo}} \leftarrow \text{clip}(\lambda_{\text{topo}}, \lambda_{\text{topo}}^{\max})$
 **end if**
 Compute loss $L(\theta, \phi) \leftarrow \lambda_{\text{rec}}(L_{\text{rec}} - \xi_{\text{rec}}) + \lambda_{\text{topo}}(L_{\text{topo}} - \xi_{\text{topo}})$
 **if** ¬InitialPhaseKL **then**
   Compute $L(\theta, \phi) \leftarrow L(\theta, \phi) + \text{KL}[q_\phi(Z_b; X_b) \,\|\, p_\theta(Z_b)]$ (batch average)
 **end if**
 update $(\theta, \phi)$ using $(\partial_\theta L(\theta, \phi), \partial_\phi L(\theta, \phi))$
 $t \leftarrow t + 1$
**end while**

---

## A.8 ARCHITECTURES

Table 14: AE (or VAE) architectures. FC represents a fully-connected layer. Conv2D and Conv2DT are a two-D convolution layer and a transposed two-D convolution layer, respectively.

| DATASET | ARCHITECTURE | |
|---|---|---|
| SWISS ROLL | INPUT | 3 |
| | LATENTS | 2 |
| | $f_\phi, q_\phi(\mathbf{z}|\mathbf{x})$ | (FC 512, RELU)×3 LAYERS |
| | $g_\theta, p_\theta(\mathbf{x}|\mathbf{z})$ | (FC 512, RELU) ×2 LAYERS |
| HUMAN MOTION | INPUT | 50 |
| | LATENTS | 2 |
| | $f_\phi, q_\phi(\mathbf{z}|\mathbf{x})$ | (FC 512, RELU)×3 LAYERS |
| | $g_\theta, p_\theta(\mathbf{x}|\mathbf{z})$ | (FC 512, RELU) ×2 LAYERS |
| COIL20 | INPUT | 32×32×1 |
| | LATENTS | 2 |
| | $f_\phi, q_\phi(\mathbf{z}|\mathbf{x})$ | (FC 256, RELU) ×6 LAYERS |
| | $g_\theta, p_\theta(\mathbf{x}|\mathbf{z})$ | (FC 256, RELU) ×6 LAYERS |
| CIFAR10 | INPUT | 32×32×3 |
| | LATENTS | 128 |
| | $f_\phi, q_\phi(\mathbf{z}|\mathbf{x})$ | (CONV2D, GELU) ×5 LAYERS, FC |
| | $g_\theta, p_\theta(\mathbf{x}|\mathbf{z})$ | FC GELU, (CONV2DT, GELU, CONV2D, GELU) ×2, CONV2DT, TANH. |

As shown in Table A.8, the same encoder and generator architectures are used for all models of each dataset. Additionally, the NVP consists of six latent variables, and each variable has three hidden layers, 256 units and residual connections. To improve the performance of the realNVP, we use gradient clipping, Softsign activation function, and non-linear output layer. We use three fully connected layers with 256 units and LeakyReLU for $p_\theta(z|\epsilon)$ of VHP.

