# OpenReview forum: "Local Distance Preserving Auto-encoders using Continuous k-Nearest Neighbours Graphs"
_ICLR.cc/2023/Conference — Submitted to ICLR 2023_

### Official Review · Reviewer_1cQZ · 2022-10-20

**Confidence:** 3
**Correctness:** 4
**Technical Novelty And Significance:** 2
**Empirical Novelty And Significance:** 2
**Recommendation:** 5

**Clarity, Quality, Novelty And Reproducibility:**

As detailed above, the writing is of high quality and the method is very clear. However, the originality is limited as the proposed method essentially combines existing ideas without justifying and supporting the choices made.

**Strength And Weaknesses:**


Pros:
- Paper is well written
- Method is clear and simple
- Experiments on a number of datasets

Cons:
- Novelty is limited
- Choices not motivated or ablated
- Additional choices not explored
- Experiments are weak

The paper is very well written and easy to follow, as a result the method and its evaluation are very clear. The method is simple and found to be effective on a number of datasets.

The novelty is limited, as the method is merely a combination of existing autoencoders, existing priors and an existing nearest neighbor graph definition.

The proposed strategy is presented as a fixed recipe, without motivating why the particular choices are made and without ablating either the algorithmic choices or the hyperparameters. The later must have been tuned but no results are given or discussed. This is important because it is unclear if hyperparameters have been also tuned for competitors. The choice of $k=9$ is given as an arbitrary choice, while for the length scale $\delta$ not even a value is given.

Why is the topological distance minimized while the reconstruction loss posed as a constraint? Why not vice versa? Why not minimize a function of both?

More importantly, why use CkNN and why this way? Using CkNN is claimed as a main contribution ("we are the first to adapt it to..."). Fixing all other choices made, how would the behavior be affected by any other graph definition? For example:
- CkNN uses the geometric mean of the kNN distances of x and y in defining an edge between x and y. Why not experiment with the generalized (power) mean, which subsumes the minimum/maximum, the RMS as well as the harmonic, geometric and arithmetic mean among others? Why not any other symmetric function?
- If the graph definition is so important, why not a weighted graph, which would result in weighted loss terms in eq. (1)? Again, a number of functions could be candidate for the weights.

Experiments are weak. In particular:
- The datasets used are small (in fact, toy) and the metrics are near perfect if I am not mistaken (assuming perfect value 0 for two of the metrics and 1 for the other two). As a result, the differences between methods are tiny.
- Tables use bold black and bold blue, which are not explained in the captions. Bold black must be the best but bold blue is unclear. This is misleading because bold blue is chosen for the proposed method.
- In Table 5 (COIL), SNE is best under 2 of the 4 metrics and in Tables 1-4 (MOCAP) the difference is 0.1%.
- In Fig. 9 (MOCAP), the latent space of UMAP looks excellent, while in most other cases UMAP is missing from the comparison with no explanation.
- On Swiss Roll and CIFAR10, metrics are perfect and there is no benefit over VR.

I would expect experiments on more realistic datasets used for unsupervised metric learning, for example CUB, Cars, SOP and InShop. These experiments should also use larger network architectures and include comparison with modern unsupervised metric learning methods.

**Summary Of The Paper:**

This work proposes a manifold learning strategy based on autoencoders (with a Euclidean distance reconstruction loss) and continuous k-nearest neighbor graphs (CkNN) (with a topological / local distance preservation loss between input and latent space). It implements a constrained optimization where topological loss is minimized such that the reconstruction loss is upper bounded.

**Summary Of The Review:**

This is a simple and interesting manifold learning strategy that makes sense but novelty is limited, choices appear to be arbitrary, additional choices are not explored and experimental evaluation is weak.

---

> ### Author Response · Authors · 2022-11-14
> **Authors’ response to Reviewer 1cQZ**
>
> General responses:
> - "The novelty is limited ...":  See general comments and response to Reviewer-n97h.
> - "The proposed strategy": In Section 2.2 we report our experiences and the best-performing choice of model & constraint combination. We discuss in detail what the problem formulation choices are and in what way the choices we make correspond to other methods  (see P2, L8-9). We address the advantages of Lagrangian optimisation vs summing weighted losses (see P2, L9-11). We also detail why we chose the constraints the way we did in  2a-b and 3a-c (see P3, L11-13). We are happy to answer any further specific questions the reviewer has that are not already addressed in 2.2. Concerning the model & topological regulariser combinations and hyper-parameters: We have 18 combinations of resulting models. The hyper-parameter choices are described in Section 3 in the dedicated paragraphs "Models" and "Hyperparameters" and again, in more detail, in the relevant sections corresponding to each experiment in the Appendix. It is a large amount of text data given all the models and the data sets we have. We will add all optimal hyper-parameters to the published code. We used the same architectures for all models and tried the same hyper-parameters in the hyper-parameter search as much as it was possible because of specific differences in the models.
>
> * "More importantly ...". The CkNN's advantages are described in detail in Section 2.4
>     * "CkNN uses the geometric mean ..." We are delighted about the reviewer's insightful comments about taking other generalised  (power) means in the graph construction and would be interested in further thoughts about what topological concepts, say, the harmonic mean corresponds to. In our view the "geometric mean" in CkNN results from or happens to correspond to topological considerations, as summarised in 2.4 and shown in Berry & Sauer (2019), it has more in-depth connections than just taking a "geometric mean" or any symmetric function for that matter.
>     * "If the graph definition is so important ..": Some methods like SNE and UMAP use weighted graphs, VR and CkNN use non-weighted graphs. It all depends on the proposed initial assumptions w.r.t. topology preservation the method is based on. We are happy to answer any specific questions w.rt. the assumptions/choices these methods make and why  they chose/propose the corresponding graph construction.
> - "I would expect experiments ...": We thank the reviewer for proposing these datasets, in this paper we chose datasets that we encountered in the related literature of manifold learning but we are happy to explore and test our method on datasets from other parts of the ML literature, however, the mentioned metric learning is not the scope of this paper.
>
>
>
> Experiments:
>
>
> - "The datasets used ...": The datasets we use are all standard datasets in the relevant literature, please consult Tenenbaum et al. (2000). McInnes et al. (2018), Li et al. (2021).
> - “Tables use bold black and bold blue …”: As we wrote in appendix A1, bold and blue bold indicate the best results in the AE-based and VAE-based models, respectively. We will move this to the main paper.
> - “In Table 5 (COIL), SNE is best …”: We would also like to highlight that when comparing methods we use evaluation metrics already used in other papers in the literature and that we strived to do a fair hyper-parameter search for all models. The metrics we use are standard in the literature for quantitative comparison. These metrics, however, indeed have their limitations because good quantitative results do not always correspond to good qualitative results, see for example UMAP-AE vs CkNN-AE or VR-AE vs CkNN-AE in Fig. 3. That is why most papers in the literature also use 2D and 3D visualisations for qualitative assessment and evaluation. For this reason, we also define another metric (distance correlation) and add qualitative comparisons such as the 2D and 3D visualisation of the latent space. The latter is a commonly accepted method for qualitative evaluation in manifold learning. It is a commonly accepted method for qualitative evaluation in manifold learning (e.g., ISOMAP [Tenenbaum et al., 2000], TOPO-AE [Moor et al., 2020]). In the visualisation of latent spaces, some of our models are better than, say, SNE-based and UMAP (e.g., Fig. 3 latent space of Swiss-roll). An additional advantage of CkNN is that it seems to be significantly faster to compute when compared to VR.
> - “In Fig. 9 (MOCAP)…”: see above paragraph. In Tables 1-4, Tables 7-11 and Fig. 3, our model outperformed UMAP-AE.
> - “On Swiss Roll and CIFAR10 .. ”: see above paragraph.

---

> > ### Comment · Reviewer_1cQZ · 2022-11-23
> > **Thank you for the feedback**
> >
> > I appreciate the authors' feedback.
> >
> > I am afraid that my major concern on novelty has not been addressed, as the proposed approach merely combines three existing elements in one solution. For this reason I have to keep my recommendation.
> >
> > My remaining concerns are also not fully addressed. In particular:
> > - the choice of constrained optimization only refers to previous works; it would be more convincing to provide an ablation in the particular experimental setting.
> > - on hyperparameters, I cannot find ablation experiments in the main paper or in the appendix.
> > - toy datasets: I understand that these may be standard choices in many papers on the subject. This does not mean that they are good choices. My suggestion to use datasets from metric learning precisely allows experiments at larger scale and evaluation against semantic classes. I do not see how this is out of scope.
> > - bold black, bold blue: Please make sure that the definition of such visual elements and styles is in every caption.

---

### Official Review · Reviewer_bRM4 · 2022-10-22

**Confidence:** 4
**Correctness:** 3
**Technical Novelty And Significance:** 2
**Empirical Novelty And Significance:** 2
**Recommendation:** 3

**Clarity, Quality, Novelty And Reproducibility:**

\The constrained optimization for generative models seems well-studied and the idea of local preserving seems widely used.

Although this paper may emphasize that the preserving mechanism is built on *distance*, it not a significant contribution for me.

The constrained optimization for generative models is also well-studied.

Another key is the utilization of continuous k-nearest neighbors, which may be not a remarkable contribution of this paper.

**Strength And Weaknesses:**

## Strength

- The paper is easy to follow.
- The paper seems technically sound and the authors discuss different situations with diverse prior distributions.
- The experiments are sufficient for me.

## Weakness
- My main concern is that the novelty seems not sufficient for me. The constrained optimization for generative models seems well-studied and the idea of local preserving seems widely used. Although this paper may emphasize that the preserving mechanism is built on *distance*, it not a significant contribution for me. Moreover, another key is the utilization of continuous k-nearest neighbors, which may be not a remarkable contribution of this paper.
- In problem (1), the loss only considers the overlapped links. Should the different links between two graphs be also important for training auto-encoder? Could the author provide some explanations?

**Summary Of The Paper:**

This paper proposes an extension of auto-encoder via incorporating the local distance preserving. Specifically, the local distance preserving is implemented by:

(1) constructing two graphs (via CKNN) on the raw features and the learned representations;
(2) comparing the two graphs via the pairs contained in both two graphs.
(3) further proposing to optimize a contrained optimization problem for the generative models (e.g., VAE).

The authors also discuss several cases with different priors.

Sufficient experiments are also reported in both main paper and appendix.


**Summary Of The Review:**

The main concern is the novelty and some questions about some parts.

I'd like to update my score after the discussion period.

---

> ### Author Response · Authors · 2022-11-14
> **Authors’ response to Reviewer bRM4**
>
> Weaknesses
> - “My main concern is ”: See response to all reviewers above.
> - “In problem (1), the loss only considers the overlapped links …”: We are afraid there is some misunderstanding here: the loss does not only consider overlapping links, it considers the union of all edges see Eqn (1). Minimising the loss corresponds to matching both edges (graphs) and distances corresponding to edges.

---

### Official Review · Reviewer_n97h · 2022-10-25

**Confidence:** 4
**Clarity, Quality, Novelty And Reproducibility:** The novelty is limited and some detai…
**Correctness:** 2
**Technical Novelty And Significance:** 2
**Empirical Novelty And Significance:** 2
**Recommendation:** 3

**Strength And Weaknesses:**

Strength: The paper is well written and easy to follow.
Weaknesses:
1. The novelty of the proposed method is limited, the proposed method just integrates the previous  local distance-preserving loss into auto-encoders;
2. In the experiments, the detailed parameter settings of auto-encoders are not clearly introduced.

**Summary Of The Paper:**

This paper introduces several auto-encoder models that preserve local distances when mapping from the data space to the latent
space. The proposed models use a local distance-preserving loss that is based on the continuous k-nearest neighbours graph which is known to capture topological features at all scales simultaneously. In order to improve training performance, the models formulate learning
as a constraint optimisation problem with local distance preservation as the main objective and reconstruction accuracy as a constraint. The proposed method provides state-ofthe-art or comparable performance across several standard datasets and evaluation metrics.

**Summary Of The Review:**

Based on the limited novelty and unclear presentation, I am willing to reject this paper.

---

> ### Author Response · Authors · 2022-11-14
> **Authors’ response to Reviewer n97h**
>
> Strengths and weaknesses:
> - “The novelty of the proposed …”: See response to all reviewers above.
> - “In the experiments …”: The hyper-parameter choices are described in Section 3 in the dedicated paragraphs "Models" and "Hyperparameters" and again, in more detail, in the relevant sections corresponding to each experiment in the Appendix. We would be happy to answer any further specific questions the reviewer has besides this general one. Since it is a large amount of text data given all the models and the data sets we have, we will add all optimal hyper-parameters to the published code. We used the same architectures for all models and tried the same hyper-parameters in the hyper-parameter search as much as it was possible due to different model parameterisations.

---

### Author Response · Authors · 2022-11-14
**General comments**

We would like to thank the reviewers for their positive criticism and insightful comments/questions. We would first like to address the common concerns. Please find the replies to each specific question in the corresponding "Official Comments".

Firstly, we would like to reiterate the main contributions of this paper:
- we introduce CkNN as a graph construction method for local distance-preserving regularisers,
- we extend local distance-preserving regularisers/constraints to hierarchical generative models,
- we extend constrained optimisation-based training methods to this general class of AE/hierarchical-VAE models thus having a single training framework for a large class of models with various topological regularisers.

Secondly, we would like to emphasise that the goal of this paper w.r.t. CkNN graph construction is to show that it works in the context of stochastic batch gradient learning in AE/VAE models, and highlight its advantages: a good balance between incentivising topological consistency and being fast/easy to compute.

We would like to reiterate that the contribution of most referenced papers in the literature concerns the graph construction method which postulates what topological properties should be conserved. In a large number of cited previous works the graph construction method is based on known previously studied topological concepts, just as in the case of this paper—see, e.g., Sammon (1969), Hinton & Roweis (2002), McInnes et al. (2018), Moor et al. (2020).

---

### Decision · Program_Chairs · 2023-01-20

**Decision:**

Reject

**Justification For Why Not Higher Score:**

All reviewers felt that the paper was not ready for publication, even after the author rebuttal.

**Justification For Why Not Lower Score:**

N/A

**Metareview: Summary, Strengths And Weaknesses:**

Thanks for your submission to ICLR.  The reviewers were in agreement that this paper is not ready for publication.  They noted several weaknesses in the paper.  Most notably, they all felt that the novelty was not high enough, though the reviewers also commented on various concerns about the experimental results as well.  On the positive side, the reviewers all appreciated the clarity of the writing, and some reviewers felt that the experiments were sufficient.